# Above-room-temperature strong intrinsic ferromagnetism in 2D van der Waals Fe₃GaTe₂ with large perpendicular magnetic anisotropy

Gaojie Zhang[1,2,5], Fei Guo[3,5], Hao Wu[1,2], Xiaokun Wen[1,2], Li Yang[1,2], Wen Jin[1,2], Wenfeng Zhang[1,2,4] & Haixin Chang [1,2,3,4] ✉

The absence of two-dimensional (2D) van der Waals (vdW) ferromagnetic crystals with both above-room-temperature strong intrinsic ferromagnetism and large perpendicular magnetic anisotropy (PMA) severely hinders practical applications of 2D vdW crystals in next-generation low-power magnetoelectronic and spintronic devices. Here, we report a vdW intrinsic ferromagnetic crystal Fe₃GaTe₂ that exhibits record-high above-room-temperature Curie temperature ($T_c$, ~350-380 K) for known 2D vdW intrinsic ferromagnets, high saturation magnetic moment (40.11 emu/g), large PMA energy density ($\sim 4.79 \times 10^5$ J/m³), and large anomalous Hall angle (3%) at room temperature. Such large room-temperature PMA is better than conventional widely-used ferromagnetic films like CoFeB, and one order of magnitude larger than known 2D vdW intrinsic ferromagnets. Room-temperature thickness and angle-dependent anomalous Hall devices and direct magnetic domains imaging based on Fe₃GaTe₂ nanosheet have been realized. This work provides an avenue for room-temperature 2D ferromagnetism, electrical control of 2D ferromagnetism and promote the practical applications of 2D-vdW-integrated spintronic devices.

Above-room-temperature intrinsic ferromagnetism with robust large perpendicular magnetic anisotropy (PMA) is critical for many magnetoelectronic devices such as magnetic tunnel junctions and magnetic random access memory[1,2]. Intrinsic ferromagnetic two-dimensional (2D) van der Waals (vdW) materials have promote the development of various multifunctional spintronic devices such as spin tunnel field-effect transistor, electron tunneling junction, giant tunneling magnetoresistance device and single-spin microscopy[3–6]. Typically, long-range ferromagnetism in 2D isotropic systems is

vulnerable for the thermal fluctuations according to the Mermin–Wagner theorem[7], but can be stabilized by the spin-wave excitation gap caused by magnetic anisotropy[8]. However, up to now, robust large room-temperature PMA only exists in conventional non-vdW ferromagnetic thin films like CoFeB (magnetic anisotropy energy density $K_u$, $2.1 \times 10^5$ J/m³)[1]. No intrinsic 2D vdW ferromagnetic crystals such as CrI₃, Cr₂Ge₂Te₆, Fe₃GeTe₂ and CrTe₂ combine above-room-temperature intrinsic strong ferromagnetism (e.g., saturation magnetic moment $M_{sat}$ only ~12.5 emu/g in CrTe₂ at

¹Center for Joining and Electronic Packaging, State Key Laboratory of Material Processing and Die & Mold Technology, School of Materials Science and Engineering, Huazhong University of Science and Technology, Wuhan 430074, China. ²Institute for Quantum Science and Engineering, Huazhong University of Science and Technology, Wuhan 430074, China. ³Liuzhou Key Lab of New-Energy Vehicle Lithium Battery, School of Microelectronics and Materials Engineering, Guangxi University of Science and Technology, Liuzhou 545006, China. ⁴Shenzhen R&D Center of Huazhong University of Science and Technology (HUST), Shenzhen 518000, China. ⁵These authors contributed equally: Gaojie Zhang, Fei Guo. ✉e-mail: hxchang@hust.edu.cn

**Fig. 1 | Crystal characterization of the vdW layered Fe₃GaTe₂ single crystals.**
**a** Front (left) and top view (right) of the crystal structure of Fe₃GaTe₂.
**b** Experimental and theoretical XRD patterns of the Fe₃GaTe₂ bulk crystal. The right panel exhibits the full width at half maximum (FWHM) of the (002) diffraction peak. Inset shows an optical image of a typical bulk crystal. **c** AFM topography of a representative mechanically exfoliated atomically-thin 2D Fe₃GaTe₂ nanosheet on a SiO₂/Si substrate. **d–f** Dark-field image, corresponding EDS spectra and elements mapping image of a Fe₃GaTe₂ nanosheet. **g**, **h** HRTEM image and corresponded SAED pattern of the Fe₃GaTe₂ nanosheet.

300 K[9]) and robust large room-temperature PMA (e.g., $K_u$ only $4.9 \times 10^4$ J/m³ in 2D CrTe₂ thin film at 300 K[10], one order of magnitude lower than widely-used CoFeB thin film)[8, 11–15]. 2D vdW intrinsic ferromagnetic crystals with above-room-temperature Curie temperature ($T_C$) and robust large room-temperature PMA are still elusive, but fundamentally important for room-temperature electrically control ferromagnetism and next-generation, room-temperature-operated 2D low-power magnetoelectronic and spintronic devices.

Here, we report a 2D vdW ferromagnetic crystal Fe₃GaTe₂ which combines intrinsic above-room-temperature strong ferromagnetism and robust large PMA. The Curie temperature $T_C$ of Fe₃GaTe₂ 2D crystals is up to ~350–380 K, record-high for known intrinsic 2D vdW ferromagnetic crystals. At room temperature, high saturation magnetic moment ($M_{sat}$, 40.11 emu/g), robust large PMA energy density ($K_u$, ~4.79 × 10⁵ J/m³ for bulk crystal and ~3.88 × 10⁵ J/m³ for 2D crystal) and large anomalous Hall angle ($\theta_{AH}$, 3%) are identified by magnetization and magneto-transport measurements. Furthermore, room-temperature thickness- and angle-dependent anomalous Hall devices and direct magnetic domains imaging based on Fe₃GaTe₂ nanosheet have been realized. This work introduces an excellent above-room-temperature 2D intrinsic ferromagnetic crystal candidate for next-generation 2D-vdW-integrated magnetoelectronic and spintronic devices.

## Results

### Characterizations of vdW Fe₃GaTe₂ crystals

Plate-like vdW ferromagnetic crystal Fe₃GaTe₂, with typical lateral sizes up to ~2 × 3 mm (inset in Fig. 1b), is grown by the self-flux method (see Methods). Fe₃GaTe₂ has hexagonal structure of space group P6₃/mmc ($a = b = 3.9860$ Å, $c = 16.2290$ Å, $\alpha = \beta = 90°$, $\gamma = 120°$). In Fe₃GaTe₂ crystal, the Fe₃Ga heterometallic slab is sandwiched between two Te layers, and the vdW gap is between two adjacent Te atoms. The slabs of Fe₃GaTe₂ are stacked along the c axis with the interlayer spacing ~0.78 nm (Fig. 1a). As shown in Fig. 1b, the X-ray diffraction (XRD) of as-grown Fe₃GaTe₂ bulk crystal shows typical (00 l) orientation with narrow full width at half maximum (FWHM, 0.03°), suggesting the strict orientation growth and high crystallinity. Figure 1c shows an atomically-thin Fe₃GaTe₂ few-layer nanosheet with 2.7 nm (3 layers) thickness isolated on SiO₂/Si substrate by Scotch tape. The triangular shape with 60° 2D crystal outline are common in some 2D crystals of hexagonal structures[16]. Moreover, Raman spectra for Fe₃GaTe₂ nanosheets with different thickness all only exhibit lattice vibration modes $A_1$ (~128 cm⁻¹) and $E_2$ (~144 cm⁻¹) of Te atoms (Supplementary Fig. 1), which are usually observed in other 2D telluride compounds[17]. Compared with the typical Raman spectra of Te thin film ($A_1$ at ~122 cm⁻¹ and $E_2$ at ~142 cm⁻¹), the slightly right-shift of $A_1$ and $E_2$ may attribute to the uniaxial strains result from the introduction of Fe₃Ga heterometallic slab[18].

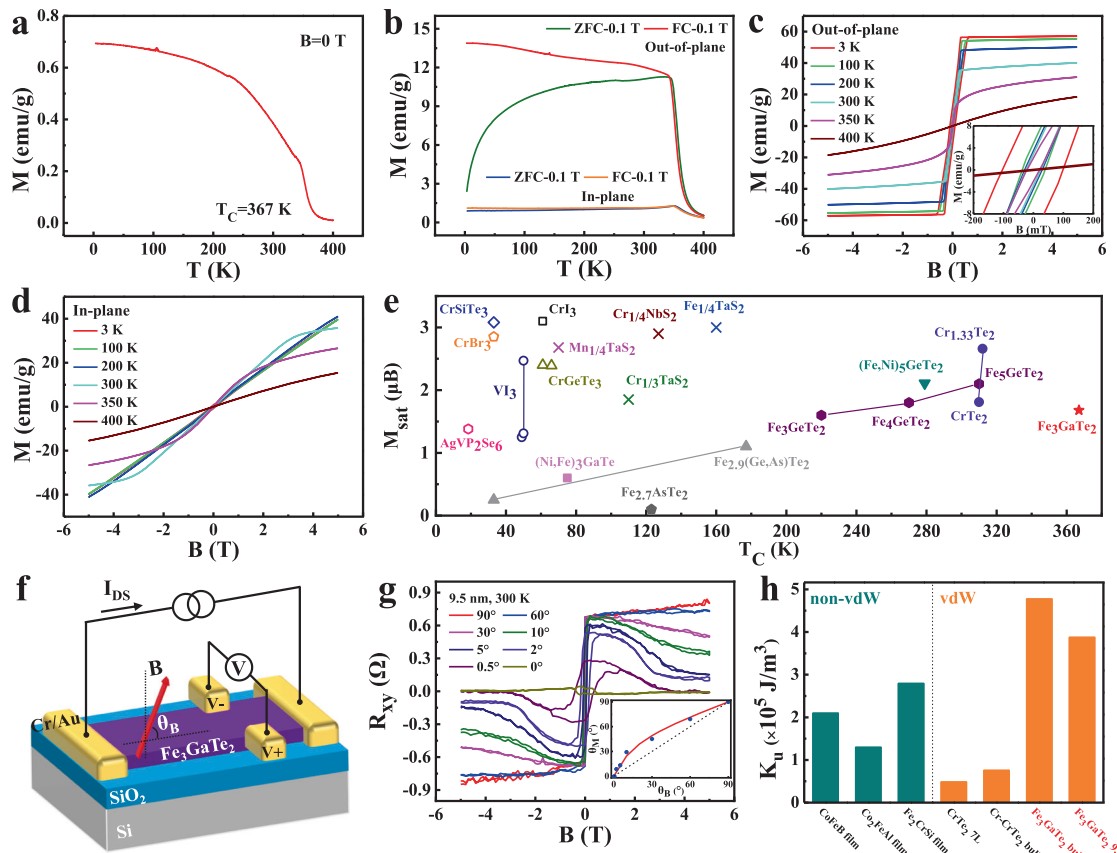

**Fig. 2 | Magnetization measurements of bulk Fe₃GaTe₂ and angle-dependent anomalous Hall effect in few-layer Fe₃GaTe₂.** Temperature-dependent spontaneous magnetization ($B = 0$) (**a**) and ZFC-FC ($B = 0.1$ T, out-of-plane and in-plane) (**b**) curves of Fe₃GaTe₂ bulk crystals. M-H curves of Fe₃GaTe₂ bulk crystals at varying temperatures with magnetic fields along the out-of-plane (**c**) and in-plane (**d**) direction. Inset in **c** shows the enlarged image of hysteresis loops. **e** $T_C$ and $M_{sat}$ comparison for various vdW-type ferromagnets, including vdW ferromagnetic insulators (open symbols), vdW ferromagnetic metals (solid symbols), and the ferromagnetic TMDC compounds (crosses) with intercalation of magnetic

transition metal atoms (M = Fe, Cr, and Mn). See reported data and references in Supplementary Table 3. **f** Schematic and measurement geometry of the few-layer Fe₃GaTe₂ Hall device. **g** Angle-dependent Hall resistance ($R_{xy}$) of a Fe₃GaTe₂ few-layer nanosheet (9.5 nm) at 300 K. Inset shows the $\theta_M$ as a function of $\theta_B$. The solid line is the fitting curve, and the dash line marks $\theta_M = \theta_B$ that corresponds to $K_u = 0$. **h** $K_u$ comparison for some conventional PMA ferromagnetic film and vdW ferromagnetic crystals at 300 K. See reported data and references in Supplementary Table 4.

Figure 1d, e and Supplementary Fig. 2a show the dark-field field-emission transmission electron microscopy (FTEM) images and energy-dispersive X-ray spectroscopy (EDS) spectra of the Fe₃GaTe₂ nanosheets. The atomic ratio of Fe, Ga and Te are 2.94 ± 0.05:1.03 ± 0.06:1.91 ± 0.01, which very close to the 3:1:2 stoichiometry. Meanwhile, we also compare the atomic ratio of Fe, Ga and Te through XPS and obtain similar stoichiometric atomic ratios (Supplementary Table 1). Further, EDS mapping images exhibit uniform distribution of Fe, Ga and Te (Fig. 1f and Supplementary Fig. 2b). Figure 1g and Supplementary Fig. 2c show high-resolution TEM image on two different Fe₃GaTe₂ nanosheets. Clear (110) and (1$\bar{2}$0) lattice planes can be identified, which both exhibit the same interplanar spacing of ~0.206 nm owing to the six-fold rotational symmetry (Fig. 1h and Supplementary Fig. 2d). Moreover, the selected-area electron diffraction (SAED) presents single crystal nature with two-fold hexagonal diffraction spots. According to the XPS spectra, the valance states of Fe, Ga and Te in Fe₃GaTe₂ are determined to be $Fe^0$, $Fe^{3+}$, $Ga^{2-}$ and $Te^{2-}$, respectively (Supplementary Fig. 3, more discussions in Supplementary Note 1). Different from the $Fe^{2+}$ and $Fe^{3+}$ in Fe₃GeTe₂ crystals with typical bulk $T_C$ ~205 K[12], the above-room-temperature ferromagnetism in Fe₃GaTe₂ may cause from the $Fe^0$ and high-spin states $Fe^{3+}$.

## Ferromagnetic properties and room-temperature magnetic anisotropy of Fe₃GaTe₂ crystals

The ferromagnetic properties of Fe₃GaTe₂ crystals are examined by vibrating sample magnetometer (VSM) in both out-of-plane and in-plane configurations. Figure 2a shows the temperature-dependent tests (M-T) under spontaneous magnetization regime without external magnetic field ($B = 0$, see Methods). The Fe₃GaTe₂ bulk crystals exhibit a typical ferromagnetic transition behavior and $T_C$ up to ~367 K. Meanwhile, the zero-field-cooling (ZFC) and field-cooling (FC) tests under a magnetic field of 0.1 T (Fig. 2b) also show a typical ferromagnetic feature, further demonstrating the ferromagnetic nature in Fe₃GaTe₂. Specially, typical magnetic field-dependent (M-H) hysteresis loops are acquired from the out-of-plane orientation while vanish under the in-plane orientation in Fe₃GaTe₂ bulk crystals (Fig. 2c, d). The significant difference between out-of-plane and in-plane M-H curves proves the out-of-plane easy axis (Supplementary Fig. 4), and a strong PMA of the magnetization with a large $K_u$ of ~4.79 × 10⁵ J/m³ is observed at 300 K (more discussions in Supplementary Note 2). The Fe₃GaTe₂ bulk crystals also exhibit the relative large coercivity ($H_C$) 1014 Oe at 3 K and small $H_C$ 249 Oe at 300 K. Additionally, the $M_{sat}$ of Fe₃GaTe₂ bulk crystals is 57.18 emu/g (1.68 μB/Fe) at 3 K and remains 40.11 emu/g (1.18 μB/Fe) at 300 K (Supplementary Table 2). Compared with other intrinsic vdW ferromagnetic crystals such as CrI₃, Cr₂Ge₂Te₆ and

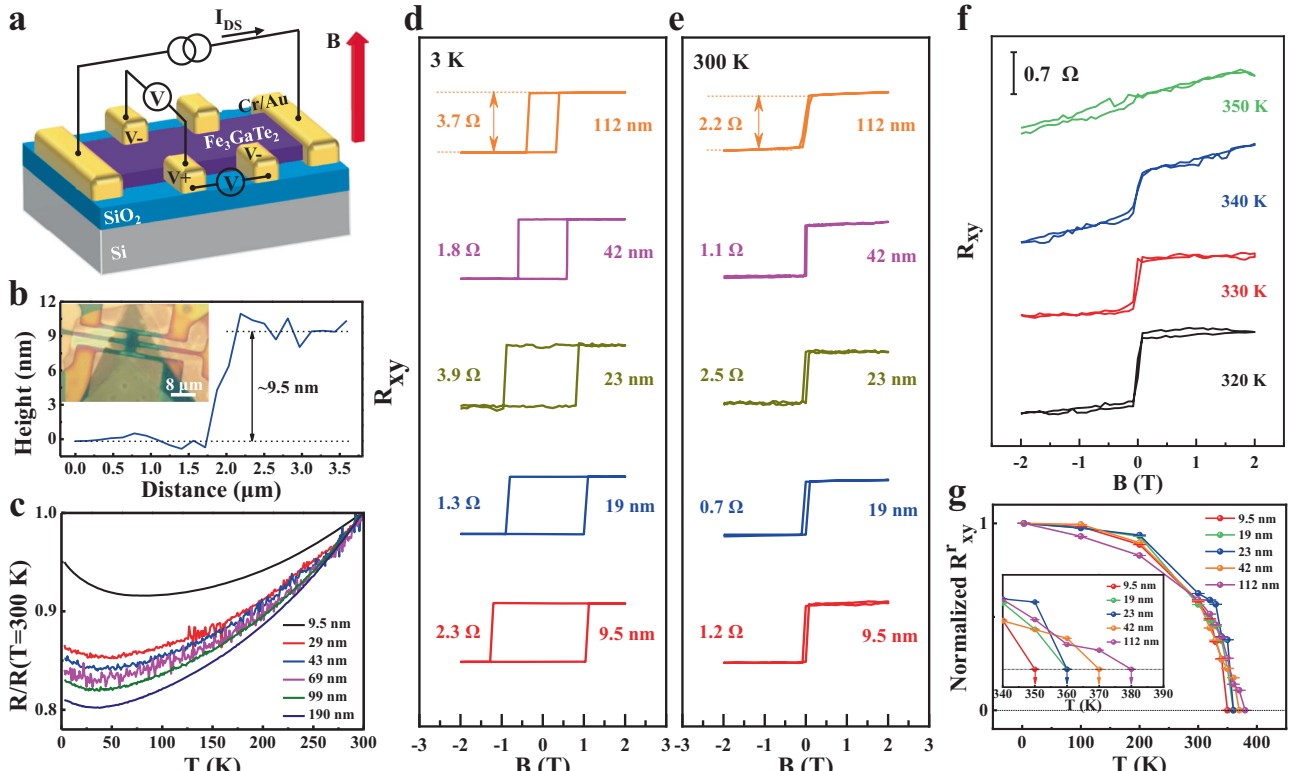

**Fig. 3 | Magneto-transport measurements and thickness-dependent anomalous Hall device performances in single-sheet Fe₃GaTe₂ nanosheets.**
**a** Schematic and measurement geometry of the Fe₃GaTe₂ Hall device. **b** Optical image and height profile of the few-layer Fe₃GaTe₂ Hall device. **c** Temperature-dependent longitudinal resistance of Fe₃GaTe₂ nanosheets with different thickness. Longitudinal resistance are normalized by their values at 300 K. Hall resistance ($R_{xy}$)
at 3 K (**d**) and at 300 K (**e**) obtained in Fe₃GaTe₂ nanosheets with different thickness. **f** High temperature (>300 K) AHE in 9.5 nm few-layer Fe₃GaTe₂. **g** Normalized remanent anomalous Hall resistance ($R^r_{xy}$) as a function of temperature in Fe₃GaTe₂ nanosheets with different thickness. $R^r_{xy}$ data are normalized by their values at 3 K. Inset shows the enlarged image from 340 to 390 K and the arrows mark the $T_C$. Error bars s.d., $N = 25$.

Fe₃GeTe₂, the Fe₃GaTe₂ combines record-high $T_C$, large room-temperature $M_{sat}$ and large room-temperature PMA (Fig. 2e and Supplementary Table 3), which is a powerful candidate for room-temperature 2D ferromagnetic platform for next-generation spintronic devices.

In addition to the large room-temperature PMA in Fe₃GaTe₂ bulk crystals, the room-temperature magnetic anisotropy of Fe₃GaTe₂ few-layer nanosheet (9.5 nm) is investigated by angle-dependent AHE (Fig. 2f). When a perpendicular magnetic field ($\theta_B = 90°$) is applied, the field-dependent anomalous Hall resistance ($R_{xy}$-B) curve presents nearly square hysteresis loop, indicating an out-of-plane magnetic anisotropy (Fig. 2g). With the decrease of $\theta_B$ from 90° to 0°, the $R_{xy}$ changes from an uptrend (for example, $\theta_B = 90°$) to a downtrend (for example, $\theta_B = 30°$) with the magnetic field in the high filed regime ($B > 1$ T), and eventually almost disappears ($\theta_B = 0°$). Since the $R_{xy}$ is only proportional to the out-of-plane component of magnetization, the results suggest that the magnetization is pinned in the out-of-plane direction at small magnetic field, and is pulled towards the in-plane direction only in the high field regime[12]. Further, the $K_u$ is estimated -3.88 × 10⁵ J/m³ by fitting the data of $\theta_B$ and $\theta_M$, where $\theta_M$ is the tilt angle between the magnetization and the sample plane (Inset in Fig. 2g, more discussions in Supplementary Note 2). Meanwhile, $\theta_M$ is always larger than $\theta_B$ which implies the magnetization is always tends to the out-of-plane direction regardless of the direction of the magnetic field, indicating a strong PMA in this few-layer Fe₃GaTe₂. The robust large $K_u$ in Fe₃GaTe₂ bulk crystals and 2D few-layer nanosheet at room temperature is not only better than the conventional widely-used ferromagnetic thin films like CoFeB and Co₂FeAl[1, 19], but also an order of magnitude larger than other vdW ferromagnetic crystals such as Cr-

CrTe₂ (7.6 × 10⁴ J/m³) and CrTe₂ (4.9 × 10⁴ J/m³)[10, 20] (Fig. 2h and Supplementary Table 4). Such robust large room-temperature PMA in Fe₃GaTe₂ will be vital to the realization of stable, compact vdW 2D heterostructures-based spintronic devices.

## Magneto-transport measurements and anomalous Hall effect (AHE) in single-sheet 2D Fe₃GaTe₂ crystals

The above-room-temperature single-sheet intrinsic ferromagnetism in 2D Fe₃GaTe₂ is revealed by AHE based on Hall devices (Fig. 3a). Figure 3b shows a representative Fe₃GaTe₂ Hall device with sample thickness identified as 9.5 nm. The paramagnetism-ferromagnetism phase transition (Supplementary Fig. 6) and metallic behavior, which resistance decreases with decreasing temperature, are observed in Fe₃GaTe₂ nanosheets (Fig. 3c). As the thickness is reduced, the Fe₃GaTe₂ nanosheets become more insulating (Fig. 3c), but room-temperature ferromagnetism still exists. Unambiguous evidence comes from the clear hysteresis in $R_{xy}$-B curves for five Fe₃GaTe₂ nanosheets (include 9.5, 19, 23, 42 and 112 nm nanosheets) at 3 K and 300 K (Fig. 3d, e). Their rectangular hysteresis loops with near-vertical jumps indicate the strong PMA and large thickness-dependent $H_C$ from 1.16 T to 0.36 T (Supplementary Fig. 7). For example, $H_C$ in 9.5 nm Fe₃GaTe₂ nanosheet is 3.2 times that in 112 nm nanosheet. Moreover, we observe the p-type conductive characteristics and evaluate the effect of thermal fluctuations on ferromagnetism by the temperature-dependent AHE in Fe₃GaTe₂ nanosheets, which is gradually weakened and eventually disappear above $T_C$ (Supplementary Fig. 8a–d). Notably, high-temperature AHE reveals the robust above-room-temperature long-range ferromagnetic order in even 9.5 nm few-layer Fe₃GaTe₂ nanosheet (Fig. 3f, additional low-temperature AHE

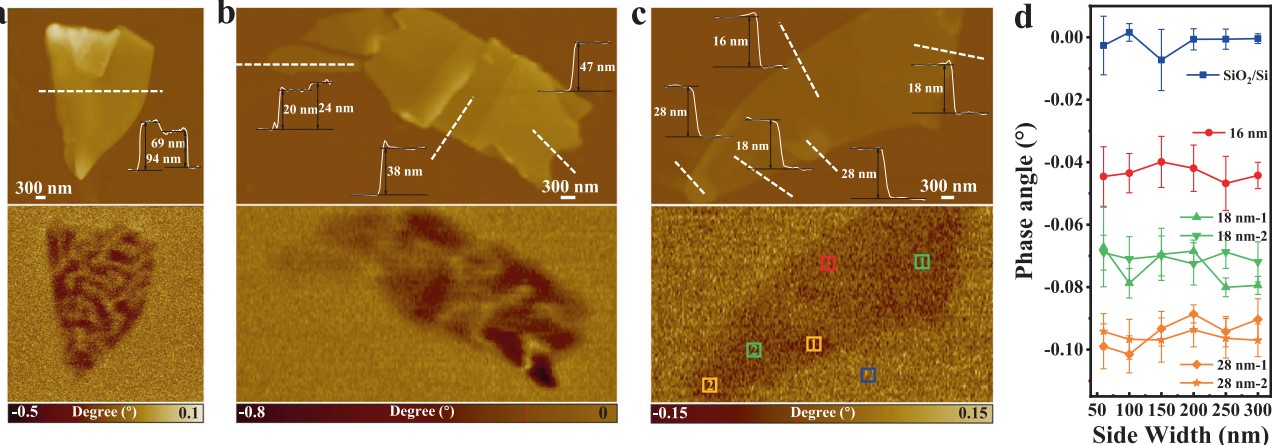

**Fig. 4 | Room-temperature direct magnetic domain imaging without external magnetic field in single-sheet Fe₃GaTe₂ nanosheets by MFM. a–c** AFM topography (top) and corresponding in situ MFM images (bottom) of Fe₃GaTe₂ nanosheets with different thickness. **d** MFM phase angles in SiO₂/Si substrate, 16 nm, 18 nm and 28 nm Fe₃GaTe₂ thin nanosheets from **c** with side width down to 60 nm. Error bars s.d., $N$ = 3.

data in Supplementary Fig. 9). The $T_C$ of Fe₃GaTe₂ nanosheets is determined by remanent Hall resistance at zero magnetic field, $R'_{xy}$ = $R_{xy}|_{B=0}$, as a function of temperature (Fig. 3 g, more discussions in Supplementary Note 3). Analysis of five Fe₃GaTe₂ nanosheets indicates that $T_C$ decreases monotonically with thickness reduced, from ~380 K for a 112 nm nanosheet to ~350 K for a 9.5 nm few-layer nanosheet, consistent with the bulk $T_C$ in VSM (more discussions in Supplementary Note 3). In any case, 2D vdW Fe₃GaTe₂ crystals exhibit an intrinsic above-room-temperature ferromagnetism with $T_C$ ~350–380 K by single-sheet AHE measurements.

In addition to the thickness effect of magneto-transport in 2D Fe₃GaTe₂, the room-temperature conductivity ($\sigma$), carrier density ($n$), mobility ($\mu$), normal Hall coefficient ($R_O$), and anomalous Hall coefficient ($R_s$) in the representative 9.5 nm Fe₃GaTe₂ few-layer nanosheet are calculated as 789.30 S cm⁻¹, 2.03 × 10²² cm⁻³, 0.24 cm² V⁻¹ s⁻¹, 3.09 × 10⁻⁴ cm³ C⁻¹, and 0.23 cm³ C⁻¹, respectively (Supplementary Table 5). With the reduction of the temperature, the carrier density monotonously increases and mobility monotonously decreases (Supplementary Fig. 10a). Furthermore, the value of $R_O$ decreases steadily with the decrease of temperature (Supplementary Fig. 10b), which is consistent with the temperature-sensitive carrier density[21]. In contrast, the $R_s$ shows a non-monotonic variation with temperature, and reaches a maximum at a certain temperature. Similar temperature dependence of $R_s$ have also been reported in ferromagnetic La₁₋ₓCaₓMnO₃, Fe₃GeTe₂, and Mn₅Si₃[21–23]. Meanwhile, $R_s$ is about three orders of magnitude larger than $R_O$, and about two orders of magnitude larger than pure Fe and Ni[24,25]. In addition, as shown in Supplementary Fig. 11, the magneto-transport measurement of a 178 nm Fe₃GaTe₂ nanosheet is presented. Combining longitudinal resistivity ($\rho_{xx}$) and anomalous Hall resistivity ($\rho_{xy}$) data (Supplementary Fig. 11a, b), the temperature-dependent anomalous Hall conductivity ($\sigma_{AH}$) and anomalous Hall angle ($\theta_{AH}$) are plotted in Supplementary Fig. 11c, d (more discussions in Supplementary Note 3). The maximum $\sigma_{AH}$ and $\theta_{AH}$ are determined as 73.25 Ω⁻¹ cm⁻¹ and 6.1 % at 3 K, respectively. Notably, the $\theta_{AH}$ persists 3% at room temperature. Such large room temperature $\theta_{AH}$ is comparable to $\theta_{AH}$ of other classical ferromagnetic materials such as Fe and (Ga, Mn)As, and much larger than that of other vdW ferromagnetic crystals[9,26,27] (Supplementary Table 6).

### Room-temperature direct magnetic domain imaging without external magnetic field in single Fe₃GaTe₂ nanosheet

The formation of magnetic domains in ferromagnets is the result of the equilibrium distribution of spontaneous magnetization satisfying the minimum energy principle[28]. In this work, magnetic domain structure and ferromagnetism of single-sheet Fe₃GaTe₂ nanosheets with different thickness are directly imaged by magnetic force microscopy (MFM) at room temperature without external magnetic field (see Methods). The strong room-temperature MFM signals compared with SiO₂/Si substrate can be detected even when the MFM scanning plane 250 nm above the surface of Fe₃GaTe₂ nanosheets, which represents an attractive interaction between Fe₃GaTe₂ and MFM tip, and indicates the strong spontaneous magnetization of Fe₃GaTe₂ 2D crystals at room temperature (Fig. 4a–c and Supplementary Fig. 12). This result provides additional persuasive evidence of room-temperature strong intrinsic ferromagnetism in the Fe₃GaTe₂ nanosheet. Notably, the Fe₃GaTe₂ nanosheets of different thickness show two kinds of magnetic domain structures: multi, stripe-domain in thicker nanosheets and single-domain in 16–28 nm thin nanosheets. The multi, stripe-domain in Fe₃GaTe₂ nanosheet resembles the stripe-domain phase in perpendicularly magnetized ferromagnetic thin films[29,30] and vdW ferromagnetic crystals Fe₃GeTe₂ and Cr₂Ge₂Te₆[31,32]. The formation of the multi, stripe-domains in Fe₃GaTe₂ implies the dominant contribution of dipolar interaction over exchange interaction and magnetic anisotropy in Fe₃GaTe₂[31]. For 16–28 nm Fe₃GaTe₂ thin nanosheets, dipole interaction gradually lose its dominance, and thus the multi, stripe-domain gradually transform to single-domain. Moreover, the MFM phase angle for the bare substrate (SiO₂/Si), 16, 18 and 28 nm thin nanosheets can be distinguished even to the side width of the selected regions down to 60 nm (Fig. 4d), suggesting the nanoscale resolution ability of ferromagnetism in MFM images of this work.

### First-principles calculations of spin-resolved electronic structures in 2D Fe₃GaTe₂ crystals

To gain more insight into the nature of the intrinsic room-temperature ferromagnetism in Fe₃GaTe₂, the spin-resolved electronic structure of bulk, trilayer and monolayer Fe₃GaTe₂ have been studied by using first-principles density functional theory (DFT) calculations (Supplementary Figs. 13 and 14). The metallic feature are presented in both spin channels of all calculated Fe₃GaTe₂ crystals, which can be seen from the two bands crossing the Fermi level together with a finite DOS at the Fermi level (Supplementary Fig. 13b). In addition, the total spin-resolved DOS shows apparent asymmetry between spin-up (↑) and spin-down (↓) channels, which is consistent with the observed intrinsic ferromagnetism in Fe₃GaTe₂. The spin-splitting of spin-up and spin-down electronic states in band structure further confirm the

intrinsic ferromagnetism (Supplementary Fig. 14a–c). For the negligible magnetic moment of Ga and Te atoms in $Fe_3GaTe_2$, the ferromagnetism mainly comes from the Fe atoms, and causes the asymmetrical electronic structure around the Fermi level. Further analysis of the Fe atom contribution reveals that its $3d$-orbital electrons mostly contribute to the Fe DOS (Supplementary Fig. 13c–e) and thus, total DOS for both spin-up and spin-down channel. Notably, the partial spin-resolved DOS for the Fe-I and Fe-II atoms are rather different, being consistent with the rather different magnetic moments on these sites (Supplementary Table 7). These differences are caused by their different local environments in crystal lattice (Supplementary Fig. 13a). With the reduction of the thickness, the metallic feature, spin-resolved DOS around Fermi level and magnetic moment of $Fe_3GaTe_2$ almost unchanged, and most contribution to DOS is still from Fe atoms and their $3d$-orbital electrons. These calculated results are consistent with the observed temperature-dependent longitudinal resistance and robust $T_C$ in 2D $Fe_3GaTe_2$ few-layer nanosheet, and stable intrinsic ferromagnetic properties caused by robust large PMA.

## Discussion

We have successfully prepared 2D vdW ferromagnetic crystal $Fe_3GaTe_2$ with both intrinsic strong above-room-temperature ferromagnetism and robust large PMA. The $Fe_3GaTe_2$ 2D ferromagnetic crystals show $T_C$ up to record-high ~380 K for known intrinsic vdW ferromagnetic crystals with high $M_{sat}$ 40.11 emu/g (1.18 $\mu B$/Fe) at 300 K. Notably, vdW $Fe_3GaTe_2$ shows robust large PMA which $K_u$ up to ~$4.79 \times 10^5 J/m^3$ for bulk crystals and ~$3.88 \times 10^5 J/m^3$ for 2D nanosheet, better than some widely-used conventional ferromagnetic films such as CoFeB and $Co_2FeAl$, and one order of magnitude larger than other 2D vdW ferromagnetic crystals. Furthermore, thickness-dependent above-room-temperature AHE in 2D vdW $Fe_3GaTe_2$ are realized with thickness down to below 10 nm. The large room-temperature anomalous Hall angle and anomalous Hall coefficient of $Fe_3GaTe_2$ nanosheets are comparable with most used traditional ferromagnetic film and much larger than known 2D vdW ferromagnetic crystals. Finally, room-temperature direct magnetic domain imaging by MFM has been realized in $Fe_3GaTe_2$ nanosheets with different thickness. This work presents a 2D vdW ferromagnetic crystal combining intrinsic above-room-temperature strong ferromagnetism and robust large PMA, and opens up new opportunities for next-generation magnetoelectronics and spintronics based on 2D vdW ferromagnetic crystals and various vdW heterostructures.

## Methods

### $Fe_3GaTe_2$ single crystal growth

High-quality $Fe_3GaTe_2$ single crystals were grown via a self-flux method. High purity Fe powders (Aladdin, 99.99%), Ga lumps (Aladdin, 99.9999%), and Te powders (Aladdin, 99.99%) in the molar ratio of 1:1:2 were placed in an evacuated quartz tube and sealed. A mixture was first heated to 1273 K within 1 h, and held for 24 h for solid reactions. Then the temperature was quickly decrease down to 1153 K within 1 h followed by slowly cooled down to 1053 K within 100 h.

### Crystal characterizations

The phase, morphology, and crystal structures of $Fe_3GaTe_2$ crystals were performed by optical microscopy (OM, MV6100), X-ray photoelectron spectroscopy (XPS, AXIS SUPRA+), powder X-ray diffraction (XRD, Smartlab SE, Rigaku Corporation; D8 ADVANCE, Brucker) with Cu Kα radiation (wavelength = 0.154 nm) and Raman spectroscopy (LabRAM HR800, Horiba Jobin-Yvon) with excitation wavelength of 532 nm. The thickness was measured by the atomic force microscopy (AFM, XE7, Park; SPM9700, Shimadzu; Dimension EDGE, Bruker). The microstructure, morphology and molar ratio were

investigated using field-emission transmission electron spectroscopy (FTEM-1, Talos F200x, FEI; FTEM-2, Tecnai G2 F30, FEI) equipped with energy-dispersive X-ray spectroscopy (EDS). All the above tests were carried out at room temperature.

### Magnetization measurements

The ferromagnetic properties of $Fe_3GaTe_2$ bulk crystals were measured by a physical property measurement system (PPMS DynaCool, Quantum Design, USA) equipped with a vibrating sample magnetometer (VSM). For the spontaneous magnetization tests, the samples were firstly heated to 400 K (above the sample's $T_C$) and hold for 5 min. Then, in order to remove the remanence of the samples and the superconducting coil, the magnetic field was raised to 2 T and decreased to 0 with oscillated mode. After that, the temperature was reduced to 3 K without external magnetic field ($B = 0$ T), and the magnetic moment test was carried out during the cooling process by VSM. For M-T curves, the rate of temperature change was set as 2 K min$^{-1}$ with the interval of 1 K. For M-H curves, the rate of magnetic field change was set as 100 Oe s$^{-1}$ with the interval of 500 Oe. 200 times per point were recorded for average in both M-T and M-H curves.

### Device fabrication and magneto-transport measurements

A standard Hall bar electrode of Cr/Au (10/20 nm) was pre-fabricated on a 300 nm oxidized layer $SiO_2$/Si substrate by using laser direct writing machine (MicroWriter ML3, DMO), e-beam evaporation (PD-500S, PDVACUUM) and lift-off. Then, mechanically exfoliated $Fe_3GaTe_2$ nanosheets were transferred onto the Hall bar pattern by using the polydimethylsiloxane (PDMS) stamp. To avoid degradation, a larger exfoliated mica flake was transferred to cover the $Fe_3GaTe_2$ nanosheets. Finally, the device is annealed in a high vacuum annealing furnace at 473 K for 1 h to form a reliable contact. All the exfoliating and transfer process were done in the Ar-filled glove box ($H_2O$, $O_2 < 0.1$ ppm).

The electrical transport properties were measured in a physical property measurement system (PPMS, DynaCool, Quantum Design) with a four-terminal configuration using silver electrodes. The magnetic field was applied perpendicular to the sample plane expect of the angle-dependent Hall resistance measurement. The resistance is tested 25 times at each temperature or magnetic field sampling point for an average with a constant current mode.

### Magnetic force microscopy measurements

The single-sheet MFM measurements were presented by the atomic force microscopy (AFM, XE7, Park) equipped with the commercial magnetic Co-Al tip (Multi75M-G, BudgetSensors). The force constant and resonance frequency of this tip are ~3 N/m and ~75 kHz, respectively. The Co-Al tip was first magnetized by a permanent magnet before the MFM measurements. MFM images were taken in a constant height mode with the scanning plane ~250 nm above its original height to eliminate the interference of short-range atomic forces. All MFM tests were performed at room temperature without external magnetic field.

### Theoretical calculation

The density functional theory (DFT)-based first-principles calculations were done by the Vienna Ab-initio Simulation Package (VASP)[33] code with the projector-augmented wave pseudopotential method. The Perdew–Burke–Ernzerhof (PBE)[34] generalized gradient approximation (GGA)[35] was performed to address the exchange and correlation effects. A slab model of the $Fe_3GaTe_2$ was adopted, and the elimination of the interactions between the slabs was achieved by a 20 Å vacuum layer along the z-axis. For trilayer $Fe_3GaTe_2$, we incorporated the long-range vdW interactions (DFT-D3 method)[36] to correct its total energy. The

Monkhorst–Pack method[37] was used for Brillouin zone sampling. The $15 \times 15 \times 3$ and $15 \times 15 \times 1$ k-point meshes were adopted to relax the bulk and monolayer (trilayer) $Fe_3GaTe_2$, respectively. We extend the wave functions to a plane-wave basis with a 400 eV cutoff energy. The energy convergence step is set as $10^{-5}$ eV. The maximum Hellmann-Feyman force after ionic relaxation is less than 0.01 eV/Å on each atom. The Gaussian smearing with smearing width of 0.1 eV was adopted for processing the partial occupancies of each wave function.

## Data availability
The data that support the findings of this study are available from the corresponding author upon reasonable request.

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

## Acknowledgements

This work was partly funded by the National Key Research and Development Program of China (No. 2016YFB0700702), the Shenzhen Science and Technology Innovation Committee (JCYJ20210324142010030), National Natural Science Foundation of China (No. 51402118, 51502101, 61674063), Guangxi Scientific Base & Talent Special Project (AD20297084) and the Fundamental Research Funds for the Central Universities (2021yjsCXCY055). The Raman, XPS and AFM tests from Analytical and Testing Center in Huazhong University of Science and Technology are acknowledged.

## Author contributions

G.Z. and F.G. contributed equally to this work. H.C. designed the project. G.Z. and H.W. prepared the materials and fabricated the devices. G.Z., X.W., L.Y. and W.J. did the crystal characterizations and physical properties tests. F.G. did the theoretical calculations. H.C., W.Z. and G.Z. analyzed the results. H.C. and G.Z. wrote the paper.

## Competing interests
The authors declare no competing interests.
