## [Peer Review File · Nature Communications]

Reviewers' Comments:

Reviewer #1:

Remarks to the Author:

The manuscript describes magnetic characteristics of vdW magnetic material, Fe₃GaTe₂. The authors synthesized Fe₃GaTe₂ for the first time via self-flux method and characterize their magnetic property from bulk to nanoscale-thickness crystal. They also measured the control of the magnetic domain wall motion by electrical gating. They attribute the controllability of domain wall motion to resultant surface charge accumulation. vdW magnet materials have received great interest due to their exotic magnetic properties, and the material the authors synthesized is the ferromagnetic material with the highest intrinsic curie temperature reported until now. As its curie temperature is higher than room temperature, which is crucial for application and commercialization, this paper is adequate to be published in Nature Communications. However, before publication, the authors need to have clear explanation for several major, and minor concerns.

Major concerns

1. From line 219 to 221, the transformation from multi, stripe-domain to single domain was observed at nanosheets whose thickness is ~20nm. From line 228 to 231, they observed single domain ferromagnetism at nanosheets with thickness 28nm, which is inconsistent with line 219~221. Consistent and clear statement is needed for which thickness transition from complex domain to single domain evolves.
2. In Fig. 4a-c, clear contrast between domains can be observed in Fig. 4a and Fig. 4b, but not in Fig. 4c. With that, authors explained it with thickness-dependent domain structure. In Fig. 4c, obvious contrast difference compared to Fig 4a, and Fig 4b cannot be observed, but still their phase angle is not uniform. How can you define the critical phase angle difference to determine whether it is single domain or multi-domain?
3. From line 233 to 246, they measured distribution of domain wall structure of nanosheets whose thickness is 145nm with increasing external electrical field. In line 236~237, they said domain wall motion 'controlled' by electric field. However, in Fig 4.g and Fig 4.h, domain wall is not controlled, only the area of domains is changed it re-distribution of domains. Conventionally, controllability of domain wall motion means the real moving of domain wall by external treatment. In this paper, their result is hardly to be said as control of domain wall motion. Should be said, for example, control of domain population via electrical gating.
4. In Fig. 4g and Fig, 4h, as gating electric field increases, phase angle was red-shifted. They observed the phenomena and concluded that it is caused by the change of magnetic anisotropy energy by the surface charge accumulation. They used thick Fe₃GaTe₂ nanosheets whose DOS at fermi level is very large, which means the system is hardly influenced by external field. Using thinner Fe₃GaTe₂ may be better to observe the phenomena.
5. From line 241 to 243, they mentioned expansion or contraction of domain wall. Basically, domain wall is boundary between domains and hardly expanded or contracted if the magnetic anisotropy is strong. Fe₃GaTe has very strong magnetic anisotropy in z-axis. I cautiously assume authors mis-stated the word domain wall. Better to revise statements.

Minor concerns

1. How can be the p-type conductivity defined in line 171 through Fig. S8a~d?
2. In imaging magnetic domain structure by MFM, the roughness of SiO₂/Si substrate can affect to the evolution of the single domain structure at thinner crystal. It's better to transfer Fe₃GaTe₂ crystal to thick hexagonal-Boron nitride (h-BN) to get exact thickness needed for evolution of single domain structure.
3. The range of electric field for gating used for controlling the domain wall motion is too narrow. To gain a better insight of the gating effect, further increase or decrease of electric field seems to be needed.

vdW magnetic materials are of high interest in condensed matter and material physics. Fe₃GaTe₂ has the highest curie temperature among them, and will be intensively researched. However, the experimental data and interpretation of it in 9th and 10th paragraph seems to be mis-approached and need more consideration to give better interpretation. I think paragraphs mentioned above is not adequate for the publication, so it is recommended to publish without them.

Reviewer #2:

Remarks to the Author:

Essentially, I think this paper should be accepted for publication in nature communications, because this contains very new and interesting idea (proposal) and experimental results of Fe_3GaTe_2 which possesses 2D layered structure with relatively high Curie temperature just beyond room temperature, leading to some applications as multifunctional spintronics devices. Device fabrications are well designed and their characterizations are highly sophisticated. I believe the technical level of this paper is extremely high and reliable. Such a propose of nano-device is very interesting: especially anomalously angle-dependent Hall devices, and gate-tunable ferromagnetic domain evolutions due to its 2D ferromagnetism. However, they should refer 2 original papers which first discovered this compounds (Fe_3GaTe_2) as room-temperature 2D ferromagnet for the first time (1) and synthesized its single crystal and discovered a large magnetic anisotropy in Fe_3GaTe_2 (2): (1) N. Kh. Abrikosov, L. A. Bagaeva, L. D. Dudkin, L. I. Pertova, and V. M. Sokolova: *Izv. Akad. SSSR, Neorg. Mater.* 21 (1985) 1680. (2) B. Chen, J. Yang, H. Wang, M. Imai, H. Ohta, C. Michioka, K. Yoshimura and M. Fang: *Journal of the Physical Society of Japan* 82 (2013) 124711. If they refer these papers, the present paper will be much better and more valuable with much reliability. They did nicely exhibit the experimental data of this material as magnetic nano-device by utilizing reliable experimental techniques and succeeded in showing interesting experiments in this field. Therefore, I recommend this paper should be accepted for publication after this modification.

Reviewer #1 (Remarks to the Author):

The manuscript describes magnetic characteristics of vdW magnetic material, Fe₃GaTe₂. The authors synthesized Fe₃GaTe₂ for the first time via self-flux method and characterize their magnetic property from bulk to nanoscale-thickness crystal. They also measured the control of the magnetic domain wall motion by electrical gating. They attribute the controllability of domain wall motion to resultant surface charge accumulation. vdW magnet materials have received great interest due to their exotic magnetic properties, and the material the authors synthesized is the ferromagnetic material with the highest intrinsic curie temperature reported until now. As its curie temperature is higher than room temperature, which is crucial for application and commercialization, this paper is adequate to be published in Nature Communications. However, before publication, the authors need to have clear explanation for several major, and minor concerns.

Reply:

Thanks reviewer for recognizing the importance and novelty of our work.

Major concerns

1. From line 219 to 221, the transformation from multi, stripe-domain to single domain was observed at nanosheets whose thickness is ~20nm. From line 228 to 231, they observed single domain ferromagnetism at nanosheets with thickness 28nm, which is inconsistent with line 219~221. Consistent and clear statement is needed for which thickness transition from complex domain to single domain evolves.

Reply:

According to the reply of Major concerns 2 and careful measurements of the thickness and phase angle difference of different regions in **Fig. 4a-c**, we consider the Fe₃GaTe₂ nanosheet of single-domain structure in this work are 16, 18, 20, 24, and 28 nm nanosheets, and the other thicker nanosheets are multi-domain structure.

2. In Fig. 4a-c, clear contrast between domains can be observed in Fig. 4a and Fig. 4b, but not in Fig. 4c. With that, authors explained it with thickness-dependent domain structure. In Fig. 4c, obvious contrast difference compared to Fig. 4a, and Fig. 4b cannot be observed, but still their phase angle is not uniform. How can you define the critical phase angle difference to determine whether it is single domain or multi-domain?

Reply:

According to our experience, the MFM imaging experiment in atmospheric environment will inevitably introduce some small disturbances from the outside environment. To avoid misjudgment, we carefully measured the average internal phase angle difference (along the white long box region, see MFM images in **Fig. R1a-c for reply only**) in each Fe₃GaTe₂ nanosheets and re-produced the **Fig. 4c, d** in revised manuscript. Please note that the internal phase angle difference of 18, 20, 24, 28 nm nanosheets are negligible, suggesting the single-domain structure. Also, the internal phase angle difference is only $\sim 0.019^\circ$ even in the typical “inhomogeneous area” of the 16 nm nanosheet, which is much smaller (~ 6 -20.5 times) than thicker nanosheets (>28 nm) with typical multi-domain structure. Therefore, we consider that the internal phase angle difference of 16-28 nm Fe₃GaTe₂ thin nanosheets are all negligible, and thus, corresponding the single-domain structure. The specific analysis is as follows:

For **Fig. R1a for reply only**, the internal phase angle difference of 94 and 69 nm nanosheets are $\sim 0.219^\circ$ and $\sim 0.114^\circ$, respectively (**Fig. R1d, e for reply only**), corresponding the typical multi-domain structure confirmed by multi-stripe MFM images. For **Fig. R1b for reply only**, the internal phase angle difference of 47 and 38 nm nanosheets are $\sim 0.39^\circ$ and $\sim 0.234^\circ$, respectively (**Fig. R1f, g for reply only**), corresponding the typical multi-domain structure confirmed by multi-stripe MFM images. Besides, please note that the internal phase angle difference of 20 and 24 nm nanosheets are negligible (**Fig. R1h for reply only**), corresponding the typical single-domain structure confirmed by non-stripe MFM images. For **Fig. R1c for reply only**,

the internal phase angle difference of 28 and 18 nm nanosheets are negligible (**Fig. R1i, j for reply only**), corresponding the typical single-domain structure confirmed by non-stripe MFM images. Besides, we also carefully measured a typical “inhomogeneous area” (see green circle in MFM image in **Fig. R1c for reply only**) inside the 16 nm nanosheet, and the phase angle difference is only $\sim 0.019^\circ$ (**Fig. R1k for reply only**), ~ 6 - 20.5 times smaller than internal phase angle difference of multi-domain nanosheets (as mentioned above, $\sim 0.219^\circ$, $\sim 0.114^\circ$, $\sim 0.39^\circ$ and $\sim 0.234^\circ$ for 94, 69, 47, 38 nm nanosheets, respectively), so we think that the 16 nm nanosheet can be considered as uniform as single domain confirmed by non-stripe MFM images.

Fig. R1 for reply only. a-c, AFM topography (top) and corresponding in-situ MFM images (bottom) of Fe_3GaTe_2 nanosheets with different thickness. d-k, Internal phase

angles difference of Fe_3GaTe_2 nanosheets with different thickness along the white long boxes in MFM images in **a-c**.

3. From line 233 to 246, they measured distribution of domain wall structure of nanosheets whose thickness is 145nm with increasing external electrical field. In line 236~237, they said domain wall motion 'controlled' by electric field. However, in Fig 4.g and Fig 4.h, domain wall is not controlled, only the area of domains is changed it re-distribution of domains. Conventionally, controllability of domain wall motion means the real moving of domain wall by external treatment. In this paper, their result is hardly to be said as control of domain wall motion. Should be said, for example, control of domain population via electrical gating.

Reply:

We agree with your statement. In this work, the gate-tunable magnetic domain evolution should be understood by the domain redistribution. Therefore, in this work, our statement that domain wall motion 'controlled' by electric field is not accurate, and control of domain population or domain redistribution by electric field may be more appropriate.

4. In Fig. 4g and Fig, 4h, as gating electric field increases, phase angle was red-shifted. They observed the phenomena and concluded that it is caused by the change of magnetic anisotropy energy by the surface charge accumulation. They used thick Fe_3GaTe_2 nanosheets whose DOS at fermi level is very large, which means the system is hardly influenced by external field. Using thinner Fe_3GaTe_2 may be better to observe the phenomena.

Reply:

As reviewer note, the van der Waals ferromagnetic Fe_3GaTe_2 crystal reported in this work is a metallic material, and its DOS at Fermi level is large, which means the system is hardly influenced by external electric field. Also, although the monolayer Fe_3GaTe_2

crystal shows smaller total DOS than that of bulk and trilayer Fe_3GaTe_2 crystals (see **Fig. S13b**), its DOS at Fermi level is still not too small, suggesting the finite effect of electric-field regulation even in monolayer crystal. Furthermore, for tests, we need nanosheet with some area, but much thinner nanosheets have too small areas to be tested. Therefore, after careful evaluation and discussion, we have decided to remove the content about electric-field-regulated domain changes, as suggested by reviewer in following comments, and we may focus on this issue in next work.

5. From line 241 to 243, they mentioned expansion or contraction of domain wall. Basically, domain wall is boundary between domains and hardly expanded or contracted if the magnetic anisotropy is strong. Fe_3GaTe has very strong magnetic anisotropy in z-axis. I cautiously assume authors mis-stated the word domain wall. Better to revise statements.

Reply:

We acknowledge that it is a mistake in our original manuscript. Domain walls are the transition regions between adjacent domains where the spontaneous magnetization gradually changes from one direction to another (*Rep. Prog. Phys.* 24 116, 1961). It is hard to make the domain walls expand or contract as the strong perpendicular magnetic anisotropy in Fe_3GaTe_2 crystal. In fact, what we want to express is that the electric field can control domain evolution rather than domain wall evolution. Therefore, in this work, we acknowledge that the description of domain wall expansion and contraction is inappropriate. As we mentioned in Major concerns 3, control of domain population or domain redistribution by electric field may be more appropriate. As we mention above, we remove this content and may focus on this issue in next work.

Minor concerns

1. How can be the p-type conductivity defined in line 171 through Fig. S8a~d?

Reply:

We follow the left-hand rule, combines the slope's sign of linear ordinary Hall effect background and the geometry of Hall effect measurement to determine the conduction type of Fe_3GaTe_2 (see **Fig. R2a for reply only**, we marked the directions of current, magnetic field and Hall voltage). Specifically, in ferromagnet, the Hall resistance R_{xy} can be regarded as the sum of ordinary Hall resistance and anomalous Hall resistance. In other words, ordinary Hall resistance appears as a linear background in the R_{xy} - B curve (see black solid line in **Fig. R2b for reply only**, positive slope of linear ordinary Hall resistance contribution are presented). Moreover, **Fig. S8a-d** also present the R_{xy} - B curves when temperature over T_c , where anomalous Hall effect disappear and only ordinary Hall effect with positive slope are shown. The conduction type can be determined from the slope's sign of ordinary Hall effect by combining the left-hand rule. In our geometry of Hall effect measurement, the positive slope represents the p-type conductivity, and thus, we define the p-type conductivity in Fe_3GaTe_2 .

Fig. R2 for reply only. **a**, Schematic and measurement geometry of the Fe_3GaTe_2 Hall device. **b**, Anomalous Hall effect in 9.5 nm few-layer Fe_3GaTe_2 nanosheet at 200 K. The linear ordinary Hall resistance contribution with positive slope are marked.

2. In imaging magnetic domain structure by MFM, the roughness of SiO_2/Si substrate can affect to the evolution of the single domain structure at thinner crystal. It's better to transfer Fe_3GaTe_2 crystal to thick hexagonal-Boron nitride (h-BN) to get exact thickness needed for evolution of single domain structure.

Reply:

In this work, the roughness of the commercial SiO₂/Si substrate we used in the MFM test is below 0.5 nm (Suzhou Research Materials Microtech Co., Ltd). Considering the thinnest nanosheet measured by MFM test in this work is a 16 nm Fe₃GaTe₂ nanosheet rather than an atomically-thin sample, we consider the effect of SiO₂/Si substrate's roughness on magnetic domain evolution is negligible. Also, please note that the temperature and electric-field tuned magnetic domain evolution of monolayer diluted ferromagnetic crystal can also present on SiO₂/Si substrate directly (*Adv. Sci.* 1903076, 2020).

3. The range of electric field for gating used for controlling the domain wall motion is too narrow. To gain a better insight of the gating effect, further increase or decrease of electric field seems to be needed.

Reply:

We agree with your statement. Further increase or decrease of electric field is expected to further influence the magnetic domain in ferromagnetic Fe₃GaTe₂ crystal. As we state in Major concerns 4, van der Waals ferromagnetic Fe₃GaTe₂ crystal reported in this work is a metallic material, and its DOS at Fermi level is large, which means the system is hardly influenced by external electric field. Even if it can be tuned slightly in the larger gate voltage, the results may still unsatisfactory, and the practical applications are limited due to more power consumption. Therefore, after careful evaluation and discussion, as you suggest, we have decided to remove the content about electric-field-regulated domain changes.

vdW magnetic materials are of high interest in condensed matter and material physics. Fe₃GaTe₂ has the highest curie temperature among them, and will be intensively researched. However, the experimental data and interpretation of it in 9th and 10th paragraph seems to be mis-approached and need more consideration to give better

interpretation. I think paragraphs mentioned above is not adequate for the publication, so it is recommended to publish without them.

Reply:

Thanks again for the reviewer's affirmation and important suggestions on this work. After careful evaluation and discussion, we have decided to accept your comments. Specifically, we decide to remove the content about electric-field-regulated domain changes, that is, **Fig. 4e-h**.

We retain other direct room-temperature magnetic domain imaging data because we believe that the presentation of magnetic domain structure is important and has reference value for the further study of 2D ferromagnetism in Fe_3GaTe_2 . At least, in this work, we have successfully achieved direct magnetic domain imaging at room temperature in Fe_3GaTe_2 ferromagnetic 2D crystals. This result provides additional persuasive evidence of intrinsic room-temperature strong ferromagnetism in Fe_3GaTe_2 nanosheet. Anyway, following your advice, we have removed some inappropriate statements, especially about electrical-field gate-tunable content. We have modified **Fig. 4** and some statements, and all changes have been marked in red in the revised manuscript.

Reviewer #2 (Remarks to the Author):

Essentially, I think this paper should be accepted for publication in nature communications, because this contains very new and interesting idea (proposal) and experimental results of Fe_3GaTe_2 which possesses 2D layered structure with relatively high Curie temperature just beyond room temperature, leading to some applications as multifunctional spintronics devices. Device fabrications are well designed and their characterizations are highly sophisticated. I believe the technical level of this paper is extremely high and reliable. Such a propose of nano-device is very interesting: especially anomalously angle-dependent Hall devices, and gate-tunable ferromagnetic domain evolutions due to its 2D ferromagnetism. However, they should refer 2 original papers which first discovered this compounds (Fe_3GaTe_2) as room-temperature 2D ferromagnet for the first time (1) and synthesized its single crystal and discovered a large magnetic anisotropy in Fe_3GaTe_2 (2): (1) N. Kh. Abrikosov, L. A. Bagaeva, L. D. Dudkin, L. I. Pertova, and V. M. Sokolova: *Izv. Akad. SSSR, Neorg. Mater.* 21 (1985) 1680. (2) B. Chen, J. Yang, H. Wang, M. Imai, H. Ohta, C. Michioka, K. Yoshimura and M. Fang: *Journal of the Physical Society of Japan* 82 (2013) 124711. If they refer these papers, the present paper will be much better and more valuable with much reliability. They did nicely exhibit the experimental data of this material as magnetic nano-device by utilizing reliable experimental techniques and succeeded in showing interesting experiments in this field. Therefore, I recommend this paper should be accepted for publication after this modification.

Reply:

Thank the reviewers for their recognition of our work and suggestions for improving the references. Please note that these two original papers mentioned by reviewer are about Fe-Ge-Te system rather than our Fe_3GaTe_2 . But they are still important for 2D ferromagnetism and we have referred them in our revision now (labeled ref. 14, 15 in revised manuscript).

Reviewers' Comments:

Reviewer #1:

Remarks to the Author:

I think the authors addressed my concerns adequately. I don't have further comments.

Reviewer #1 (Remarks to the Author):

I think the authors addressed my concerns adequately. I don't have further comments.

Reply: Thank you again for your comments and recognition of our work.